# Influence of Melt-Pool Stability in 3D Printing of NdFeB Magnets on Density and Magnetic Properties

**DOI:** 10.3390/ma13010139

**Published:** 2019-12-29

**Authors:** Mateusz Skalon, Michael Görtler, Benjamin Meier, Siegfried Arneitz, Nikolaus Urban, Stefan Mitsche, Christian Huber, Joerg Franke, Christof Sommitsch

**Affiliations:** 1IMAT Institute of Materials Science, Joining and Forming, Graz University of Technology, Kopernikusgasse 24, 8010 Graz, Austria; mateusz.skalon@tugraz.at (M.S.); benjamin.meier@joanneum.at (B.M.); christof.sommitsch@tugraz.at (C.S.); 2Joanneum Research, Materials—Institute for Laser and Plasma Technology, Leobner Straße 94, 8712 Niklasdorf, Austria; michael.goertler@joanneum.at; 3Institute for Factory Automation and Production Systems, University of Erlangen-Nürnberg, Fuerther Strasse 246b, 90429 Nuremberg, Germany; Nikolaus.Urban@faps.fau.de (N.U.); joerg.franke@faps.fau.de (J.F.); 4Institut für Elektronenmikroskopie und Nanoanalytik, Steyrergasse 17/III, 8010 Graz, Austria; smitsche@tugraz.at; 5Physics of Functional Materials, University of Vienna, 1090 Vienna, Austria; huber-c@univie.ac.at

**Keywords:** NdFeB, selective laser melting, additive manufacturing, melt track, powder thickness

## Abstract

The current work presents the results of an investigation focused on the influence of process parameters on the melt-track stability and its consequence to the sample density printed out of NdFeB powder. Commercially available powder of Nd7.5Pr0.7Fe75.4Co2.5B8.8Zr2.6Ti2.5 alloy was investigated at the angle of application in selective laser melting of permanent magnets. Using single track printing the stability of the melt pool was investigated under changing process parameters. The influence of changing laser power, scanning speed, and powder layer thickness on density, porosity structure, microstructure, phase composition, and magnetic properties were investigated. The results showed that energy density coupled with powder layer thickness plays a crucial role in melt-track stability. It was possible to manufacture magnets of both high relative density and high magnetic properties. Magnetization tests showed a significant correlation between the shape of the demagnetization curve and the layer height. While small layer heights are beneficial for sufficient magnetic properties, the remaining main parameters tend to affect the magnetic properties less. A quasi-linear correlation between the layer height and the magnetic properties remanence (J_r_), coercivity (H_cJ_) and maximum energy product ((BH)_max_) was found.

## 1. Introduction

NdFeB permanent magnets attract great interest due to their extraordinary performance in comparison to other types of hard magnets. Their energy density (e.g., magnetic polarization J_r_ higher than 1 Tesla, coercivity H_cJ_ above 1000 kA m^−1^, and figure of merit (BH)max greater than 500 kJ·m^−3^) outperforms other types of permanent magnets at room temperature. Due to this characteristic, NdFeB magnets are considered as key components for clean energy application [1]. They are also classified as potentially the best candidates for the miniaturization of electric devices or highly efficient electric motors [2,3]. Currently applied techniques, such as sintering, extrusion, or spark plasma sintering (SPS), deliver magnets with excellent magnetic properties and of high relative density [4,5,6,7,8,9,10,11,12,13,14]. Traditional manufacturing, however, limits the application of the magnets due to limits in design and specific tooling requirements for each design and limitations in shape flexibility and complexity [15,16]. With the technique of additive manufacturing it is possible to produce parts with more complex design; however, obtaining printing processes for functional materials is still a topic in research and development. Earlier research already showed that techniques such as slurry jetting or laser powder bed fusion (LPBF) proved to be capable of delivering net-shape magnets [17,18,19,20,21,22]. Slurry jetting has been shown to be capable of producing magnets with good magnetic properties (H_cJ_ = 775.3 kA/m, J_r_ = 0.478 T, and (BH)_max_ = 39.78 kJ/m^3^). These values reach nearly half of the values expected for the bulk material. This manufacturing approach allows the manufacture of the desired magnet in a time- and cost-efficient manner, however, and the resulting magnetic properties will always be lower than the bulk magnets, due to the presence of a significant quantity of bonding resin. Another approach was presented by Kolb et al. [17,18] who utilized laser powder based fusion (LPBF) technique in order to overcome the problem of low relative density. They managed to 3D print cubical magnets of 85% of relative density, which were characterized by a polarization of 0.514 T. The surprisingly lower than expected magnetic properties were assigned to the self-demagnetizing of the samples while measured in an open magnetic circuit inside a Helmholtz coil. Other researchers [19] also applied a laser powder based fusion (LPBF) process to the powdered NdFeB (Nd7.5Pr0.7Zr2.6Ti2.5-Co2.5Fe75B8.8) powder. The 3D printed magnet reached roughly 60% of a bulk counterpart at relative density 92% (Hc = 695 kA/m, J_r_ = 0.59 T, and (BH)max = 45 kJ/m^3^). The coercivity of the LPBF-printed samples can be also substantially increased by a grain boundary infiltration method [22]. The main conclusion was to increase the rare elements quantities in order to achieve further progress in 3D printed magnets. The main goal of the present investigation was fine tuning of the printing parameters in order to understand the melt-pool behavior under range of parameters and its influence on the density and magnetic properties. 

## 2. Materials and Methods

A spherical powder of Nd7.5Pr0.7Fe75.4Co2.5B8.8Zr2.6Ti2.5 (at.%) alloy manufactured by Neo Magnequench company (Singapore, Singapore) was used for the experiment (Figure 1). The powder particle size distribution was presented in Figure 1a.

The single melt tracks were manufactured by a method similar to that as presented in [7,17,18,20]. Previous authors manufactured single tracks by treating a layer of powder of constant depth with a laser beam with changing laser parameters. This time, however, single tracks were printed using a specially designed powder holder, which was placed in the printing chamber as a base plate. The holder was made of ferritic stainless steel 1.4301 and consisted of the slope plane with a total length of 50 mm and the linearly increasing powder thickness from 0 μm up to 125 μm. The slope angle was equal to 0.0172°. The shape deviation of the plate was: −0.2 μm and +0.7 μm at the shallowest and deepest points, respectively (Figure 2). The powder layer was equalized manually. The single lines were printed perpendicularly to the slope with intervals of 500 μm using the parameter sets as listed in Table 1. This approach allowed the investigation of the powder depth influence on the melt-track behavior. 

The laser beam was set as defocussed in order to reproduce the beam as it is when printing the bulk part of the component (hatching). The laser beam spot size was equal to 80 μm. The single tracks were cut perpendicularly in respect to their length and evaluated according to the scheme presented in Figure 3.

The radius was calculated as a radius of a circle described on a triangle defined by the most distant points of length and height (X, Y, Z in Figure 3a). Contact angle η was derived as an average value of η1 and η2 presented in Figure 3b. The bulk cubic samples of size 5 × 5 × 5 mm^3^ were manufactured using Farsoon FS121M (Farsoon Europe, Suttgart, Germany), the same 3D printing system also in under protective atmosphere of argon. The powder layer thickness was incrementally increased with step of 20 μm. The density of the bulk samples was measured using water displacement method (Archimedes method). Microstructures were revealed by etching polished surfaces with 0.5% Nital for 1 second. Grain size distribution was calculated using an algorithm presented by Rabbani and Ayatollahi [23]. Energy-dispersive X-ray spectroscopy (EDX spectroscopy or EDXS) was used for evaluation of the chemical composition. X-ray diffraction measurements were performed using a Siemens D5005 diffractometer (Siemens, Munich, Germany). For magnetic characterization the specimens were magnetized using an impulse magnetizing device IM-K-008020-AD-DT-C11A1H3A2A1 from Magnet Physik (Dr. Steingroever GmbH, Cologne, Germany) with a magnetizing voltage of 1600 V. The voltage generated a flux density over 3 T inside the magnetizing coil that led to full saturation of NdFeB while keeping the thermal stress of the equipment within an acceptable range. The magnetic measurements were performed inside a Brockhaus Hystograph HG 200 (Dr. Brockhaus Messtechnik GmbH & Co. KG, Lüdenscheid, Germany) with a J-compensated measuring coil TJH 10 according to IEC 60404-5. The measuring temperature was set to room temperature of 22 °C. The measuring procedure consisted of three phases: First, the saturation of the sample was checked within the first quadrant of the hysteresis loop. Subsequently the demagnetization curve was passed with constant flux change. After full demagnetization, the magnetization region in the opposite direction in the third quadrant was entered. The specimens were magnetized and measured in the building direction only.

## 3. Results and Discussion

Figure 4 shows that increasing linear energy increased the melt-track stability over the range of powder layer thicknesses. Application of low linear energy (I and II) resulted in a disturbed melt track. The greater the thickness of the powder layer, the more difficult it was to form an undisturbed melt track.

A powder melted by a laser on a plane forms a liquid semi-cylinder of a defined length. According to previous research [24], the stability of this semi-cylinder is dependent on its geometry: λ: Length of liquid cylinder, θ: Angle between vertical plane and the diameter perpendicular to the contact angle of a cylinder, D: Diameter (Figure 5). According to analysis performed by Gusarov and Smurov [24], a liquid semi-cylinder remains stable only when the following, essential condition is adequately fulfilled:(1)$$\frac{\pi D}{\lambda }>\sqrt{\frac{\theta \left(1+cos2\theta \right)-sin2\theta }{2\theta \left(2+cos2\theta \right)-3sin2\theta }}$$

In the limit of a circular cylinder θ=π gives:(2)$$\frac{\pi D}{\lambda }>\sqrt{\frac{2}{3}}$$
applicable at θ>π/2.

Transgression of boundary condition (Equation (2)) resulted in immediate loss of the cylinder stability and, furthermore in, balling (e.g., Figure 4: Set I at 100 μm). Since, in the given experiment, the melt-track length was not measured the stability conditions (Equations (2)), which are strongly dependent on the melt-track length, will not directly be further considered. However, based on these, when the stability condition (Equation (2)) was met, the melt track will remain stable independent of the molten semi-cylinder length (Figure 5b), either in nondisturbed or in disturbed (meandering) form [24]. Since the Θ angle is perpendicular to the radius curvature, Equation (2) can be further expressed as:(3)η<90°
where η is a contact angle as indicated in Figure 5a. 

Single tracks (STs) were cut perpendicularly to their length and their cross-sections (CSs) were measured as presented in the scheme (Figure 3). The amount of the powder melted by a laser beam is dependent on the powder stored in a powder layer and the energy delivered by the beam. The measured values were thus plotted in functions of the single track cross-section (STCS) area to achieve clarity (Figure 3 and Figure 6).

As presented in Figure 6a, the higher the linear energy value, the greater the STCS radius. A large radius is desired since this assures the stability of the melt track. When the linear energy applied was too low (set I), the melt-track radius had an approximately linear dependence on its area, which confirms the observation from Figure 4 of the poor stability of the melt track and a poor connection to the base plate. For the set II parameters coupled with a decreasing STCS area, the melt track tended to be wider and flatter, since the radius was increasing (Figure 6a). This observation is supported by data presented in Figure 6b where aspect ratio (AR) as a function of STCS area was presented. Independent of the powder layer, the higher the linear energy, the flatter the melt tracks. The contact angle between the melt track and the base plate also tended to decrease (improve) along the increase of the linear energy value (Figure 6c). The energy density for providing melt-track stability (sets III and IV) was found to remain in agreement with previous findings [17,18,19]. Further, the cubic samples were manufactured using parameters sets I–IV with a step-wise change of the powder layer depth (step = 20 μm) and their density was measured. It was found that the samples produced using parameters sets I and II provided samples so fragile that they crushed when detaching (off the plate operation). It was, therefore, impossible to perform further activities using sets I and II.

Density analysis showed that the porosity of produced magnets was composed of two significant constituents: (1) closed pores and (2) opened pores. The latter, however, consisted of both opened pores and cracks, which are indistinguishable by the selected method. Minimum value of closed porosity was found for parameters set III and set IV when the powder layer was set to 40 μm and 60 μm, respectively (Figure 7a,b). When this value was exceeded, then the amount of closed pores grew steadily along with the increasing thickness of the powder layer. The number of opened pores decreased steadily when parameters set III was used (Figure 7a) and remained roughly constant when parameters set IV was applied (Figure 7b). An increase of the closed porosity share, however, cannot be compensated by decreasing the share of opened pores, and the peak of the relative density was thus found for parameters sets III and IV when the powder layer thickness was set to 40 μm and 60 μm, respectively. A maximum density of 90.9% was reached for the samples printed with parameters set IV when a powder layer of 60 μm was applied. This stays in opposition to other research results [17,18,19] that reported similar findings for powder layer thickness of 20 μm. For the set of parameters III the maximum relative density was 86.5% and was recorded when powder layer thickness was equal to 40 μm. This implies that in order to reach a high relative density the laser beam parameters have to be coupled with an adequate powder layer thickness. The maximum density was limited by the challenging brittle behavior of the alloy, which was very sensitive to cracking induced by inner tensions. Increasing the laser energy to a certain extent would in theory lead to increased dimensions of the melt pool, better connection to underlying layers, and, in consequence, to a higher overall density. This was prohibited by the fact that samples produced with a higher laser energy already break during manufacturing, because of the increased inner tensions induced by the shrinking of the bigger melt pool during the cool-down phase. This effect has also been mentioned in [18] as a limit for the maximum density. For other materials that are also challenging to process by LPBF, such as ceramics, manufacturing at elevated ambient temperature by means of a powder bead heating has been proven to be a suitable countermeasure [25].

Figure 8a shows that there were no visible major shape distortions. Figure 8b highlights both spots where fusion was lacking and also the presence of cracks. Observation of cracks present in the sample matches the observation made in a previous work [19]. Thus, as shown in the images in Figure 8, samples could be printed with a high level of precision and without any major defects.

## 4. Microstructural and Phase Analysis

Grain size distribution was checked and compared on two samples with similar overall density, but different magnetic properties: IV-40 and IV-80. The investigated area was divided into 7 fields as presented in Figure 9 and a minimum of 4 photos were taken in random spots from each field. 

Grain boundaries detection and grain size analysis were performed using algorithm presented by Rabbani and Ayatollahi [23]. The analysis results along with representative microphotographs of the microstructures are presented in Figure 10. 

With the increase of the powder layer thickness, grain coarsening was observed as b10, b50, and b90 and values for IV-40 sample were 0.65 μm, 0.94 μm, and 1.56 μm, respectively, while these values for sample IV-80 were 0.79 μm, 1.28 μm, and 2.29 μm. The origin of grain coarsening with increasing layer thickness was in the longer cooling time for the melt pool. For complete consolidation, more laser energy was required, which in turn was proportional to the volume of the melt pool. The cooling time, which had a direct effect on the grain size, was in scale with the melt volume. Regardless of their size, the shape of observed grains did not change, however, and resembled polygons with no defined orientation. No microstructural differences were found between areas neighboring the weld lines and those placed between the weld lines. A comparison with the typical grain sizes of conventional manufactured magnets is given in Table 2. The grain sizes realized with LPBF were directly between the conventional processing routes. As a consequence, the microstructure can be selectively manipulated using different layer heights. Because of the heat dissipation during cooling of the melt pool through underlying layers, the already consolidated material was exposed to an indirect, short-term heat treatment, which initiated grain coarsening. The result was a minimum grain size of ~0.5 µm for both samples. With increasing layer height, the geometric dimension of the melt pool increased, and, therefore, also the amount of heat that was conducted through the neighboring area. 

The EDX analysis performed on unetched samples showed that there were neither chemical gradients nor inhomogeneity of elements distribution along the structure (EDX mappings attached in Appendix A) (Figure 11). This means there was a homogeneous alloy distribution in the middle of the melt tracks and also along the border towards neighboring tracks. Exsolutions or free iron, as [19] proposed, were not detected. Additional EDX line scans performed also on etched samples were attached as Appendix A. 

XRD analysis was performed on samples of similar overall density but different magnetic properties, e.g., IV-40 and IV-80. XRD patterns that were compared in Figure 12 showed no significant differences. This was unsuccessful, however, for identifying phase composition. The lack of differences show that no phase changes were induced while different powder layer thicknesses were applied and that the main factor influencing the magnetic properties was the grain size. 

## 5. Results, Description for the Magnetic Tests

Samples printed with a powder layer thickness of 20 μm for parameter set IV (as shown in Table 1) were crushed during processing before further tests could be performed. The same was true for samples printed with a powder layer thickness of 120 μm for parameter set III. Figure 13 shows the J/H-curve for parameter set IV dependent on the layer height. The curves for parameter set III are similar and these are available in the Appendix A.

The shape of the demagnetization curves, and, therefore, the magnetic behavior, changes with increasing layer height in their curvature from a concave form, that is similar to conventional magnets, towards a convex form. This will affect the flux density available when the magnet is in operation under a weakening field. A small layer height is thus preferred when sufficient magnetic properties are in the focus of the investigation. The origin of the curvature change was expected to correlate with the magnet microstructure. While J_r_ was mostly dependent on the amount of the hard magnetic Φ-phase within the volume, the resistance against demagnetization H_cJ_ was greatly affected by the phase distribution and homogeneity of the microstructure and the composition of the grain boundary phase [22]. In LPBF, the powder was completely melted, which resulted in the formation of a new microstructure responsible for the coercive field strength. This, in turn, resulted in significantly different values for H_cJ_ and BH(max) compared to plastic-bonded magnets produced from the initial material.

The discontinuous drop of coercivity of specimens built with 60 µm at −80 kA/m could be caused by the spontaneous widening of cracks as a result of the magnetic load. Similar behavior can be detected, when the LPBF-produced magnets were under critical mechanical load.

Figure 14 shows the relationship between the layer thickness and the magnetic key properties. In general, a nearly linear loss of magnetic properties with increasing layer height can be derived. 

The evolution of the microstructure with increasing layer height, as shown in the previous section, underlines the results from the magnetic measurements for different layer heights. An increased layer height resulted in a decrease in magnetic properties and, therefore, it can be concluded that a decrease in magnetic properties, as shown in Figure 14, was most likely caused by a coarsening in grain structure. A clear correlation between relative density and magnetic properties, as indicated in other studies [17,18], was not recognizable, e.g., sample IV-60 had lower magnetic properties (Figure 14) in spite of having higher density than IV-40 (Figure 7b). The remnant polarization J_r_ was the most robust indicator in the context of the layer height. This indicates that the amount of Φ-phase inside the material was not as significantly changed by the process as the overall microstructure. The difference between the maximum polarization of 516 mT and the theoretical maximum of the alloy of 730–760 mT [27] was partly a result of the remaining porosity of approx. 15% (sample B-40 μm). Even with 100% theoretical density, the polarization would only reach 593 mT and can therefore not fully explain the loss. Due to the lack of rare earth content within the alloy, a slight loss of Nd/Pr into a non-ferromagnetic phase can be expected. The coercivity dropped at a rate of 15%/20 µm layer height for set IV and with an even fall rate for set III. The maximum energy product, (BH)max, dropped as a result of the changed curvature in a manner similar to that for the coercivity. As J_r_ changed in a minor manner, the drop of (BH)max was mainly influenced by the drop of H_cJ_. Magnetic properties were compared with state of art in Table 3. As the sampling includes only one magnet investigated at every process point, no statistical statements regarding the derivation of the parameters or also the confidence levels can be made. 

## 6. Conclusions

A spherical powder of Nd7.5Pr0.7Fe75.4Co2.5B8.8Zr2.6Ti2.5 alloy was tested. The investigations carried out showed that it is possible to 3D print magnets with a relative density of 90.87%. It was found that the investigated material had a very narrow processing window and 20 W change in laser power may result in destabilization of a melt track. It was shown that shape factors of the melt track were strongly dependent on the powder bed depth, laser power, and laser speed. It was found that the higher the energy density, the more stable the melt track became. It was also found that magnets of high relative density may only be obtained when an adequate powder layer thickness was selected for the given printing parameters. The internal porosity of the printed magnets increased with increasing deviation from optimum powder layer thickness. Opened porosity, however, remained at a stable level when the powder thickness exceeded the optimum value and increased rapidly when the powder layer was too shallow (this was caused by excessive cracking). The investigations that were carried out revealed that the stability of a melt track in LPBF-treated Nd7.5Pr0.7Fe75.4Co2.5B8.8Zr2.6Ti2.5 powdered alloy is prone even to small changes in LPBF process parameters. It was found that the melt-track stability increased with increasing energy density (in tested range). The linear energy must be at least at level of 0.300 J mm^−2^, since lower energy inputs do not provide an adequately stable melt track. The magnetic properties and also the shape of the J/H-curve showed a significant dependency of the process parameters. With increasing layer height, a change in the shape of the J/H-curve was detected. The highest magnetic values were comparable to those published in similar publications. It was possible for the first time to match the magnetic performance of LPBF-treated magnets with their internal microstructure. Increased magnetic parameters were obtained with decreasing layer height. This effect is explained by a decreasing grain size with decreasing layer height due to the level of laser energy applied. The overall grain size was in line with the coercivity values for isotropic NdFeB. Similar to nanocrystalline magnets, a smaller grain size is beneficial for higher H_cJ_. For further improvement of the process, an adjusted alloy composition is necessary. Originally designed for grain sizes well below 1 µm, the current composition is not optimal for the resulting grain size distribution of 0.5–3 µm. It was also shown that the process parameters in the investigated range did not alter the phase composition of the LBPF printed magnets. The grain size coupled with the overall density are the main factors of influence on magnetic properties. Investigating spherical powder with a composition designed for sintering could be a promising approach, as the grain size can be adjusted after the LPBF process employing different heat treatment steps. 

## Figures and Tables

**Figure 1 materials-13-00139-f001:**
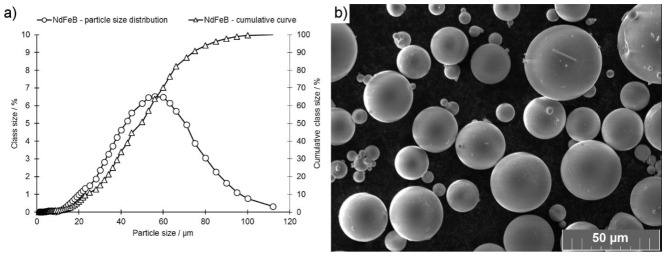
(**a**) Particle size distribution of the powder particles shown as a percentage of the total volume and cumulative with respect to increasing particle size. (**b**) SEM image of the Nd7.5Pr0.7Fe75.4Co2.5B8.8Zr2.6Ti2.5 powder used in this work.

**Figure 2 materials-13-00139-f002:**
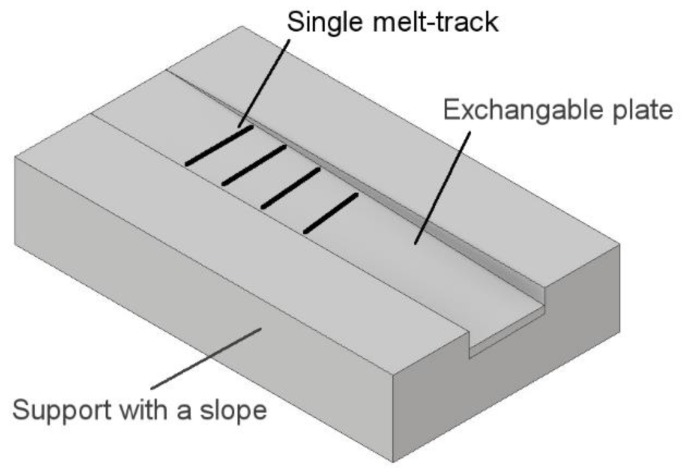
Schematics of the test stand.

**Figure 3 materials-13-00139-f003:**
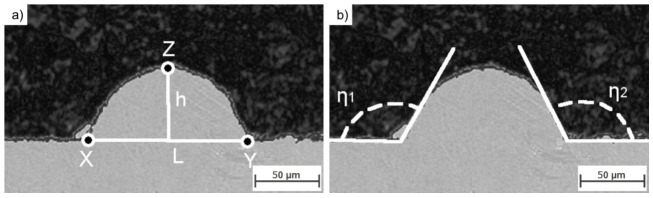
Scheme of single track cross-section (STCS) measurement: (**a**) Base length (L), height (h), contact points (X, Y), and the top point (Z); (**b**) contact angles η_1_ and η_2_.

**Figure 4 materials-13-00139-f004:**
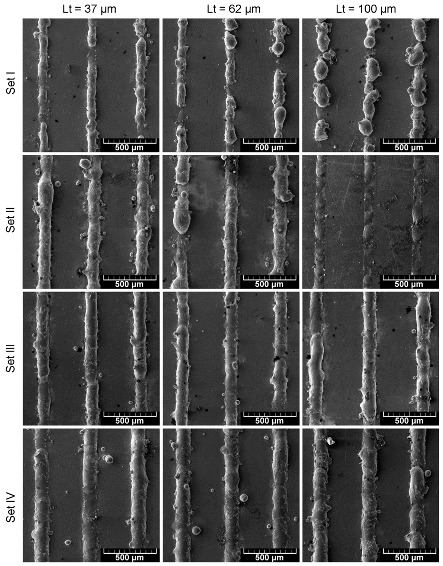
Overview of single tracks formed by parameters sets: I, II, III, and IV at various powder layer thickness (Lt).

**Figure 5 materials-13-00139-f005:**
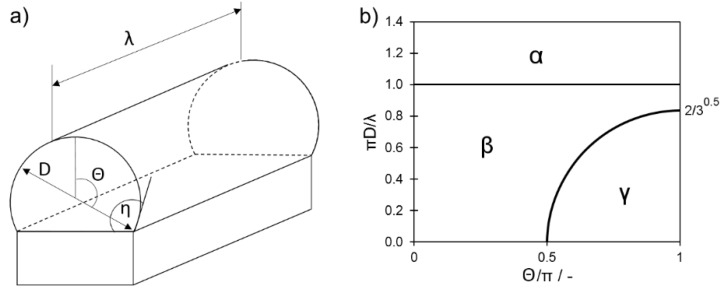
(**a**) Scheme of cylinder measurement and (**b**) map of liquid cylinder stability, where α: Both nondistorted and distorted cylinders are stable, β: Distorted cylinder is stable while nondistorted becomes unstable, and γ: Both cylinders are unstable. Based on [24].

**Figure 6 materials-13-00139-f006:**
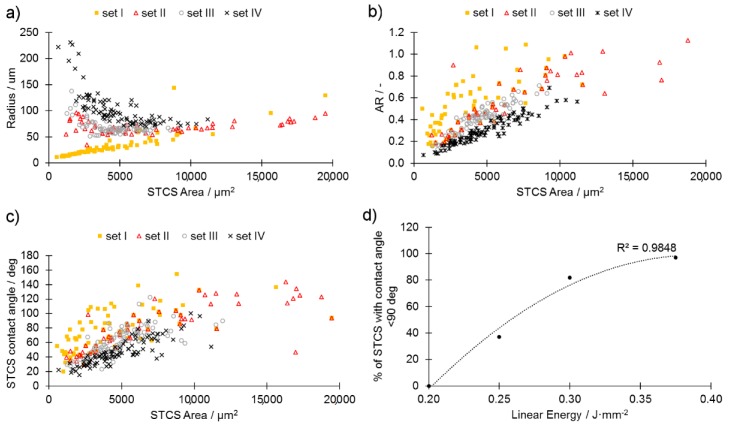
Single track cross-section (STCS): (**a**) Radius in function of STCS area, (**b**) aspect ratio (AR = h/L) in function of STCS area, (**c**) STCS contact angle in function of its area, (**d**) percentage of STCS with contact angle η < 90° in function of linear energy.

**Figure 7 materials-13-00139-f007:**
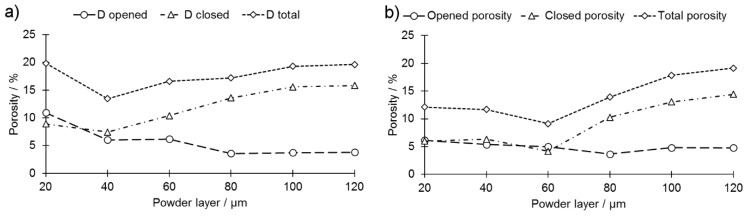
Changes of porosity structure and volume in function of a powder layer thickness: (**a**) set III (**b**) set IV.

**Figure 8 materials-13-00139-f008:**
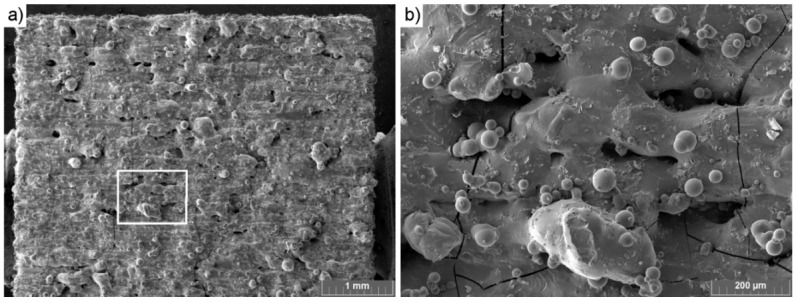
LPBF-printed NdFeB sample: (**a**) An overview, (**b**) the detail indicated in Figure 8a. (Parameters set IV, powder layer height: 60 μm).

**Figure 9 materials-13-00139-f009:**
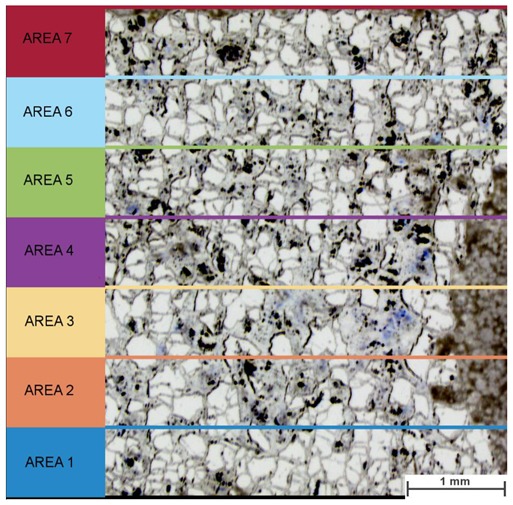
Scheme of a sample division for microstructure evaluation.

**Figure 10 materials-13-00139-f010:**
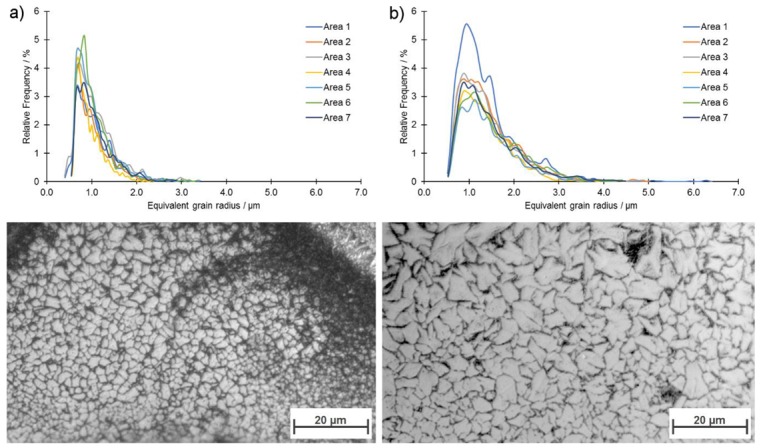
Grain size distribution with exemplary grain images of: (**a**) Sample IV-40 and (**b**) sample IV-80.

**Figure 11 materials-13-00139-f011:**
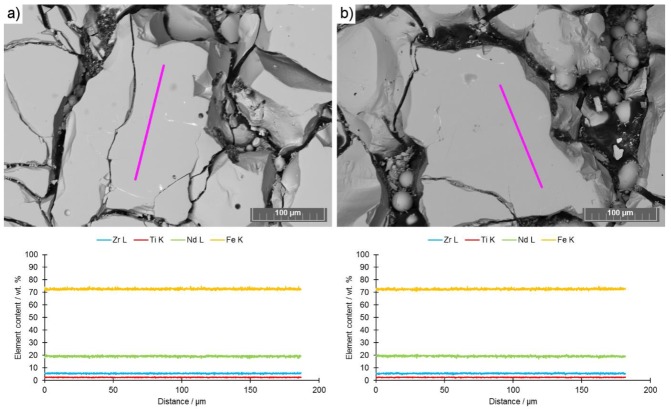
Energy-dispersive X-ray (EDX) line- scans of (**a**) samples IV-40 and (**b**) sample IV-80 with images of the spots the measurements were taken from.

**Figure 12 materials-13-00139-f012:**
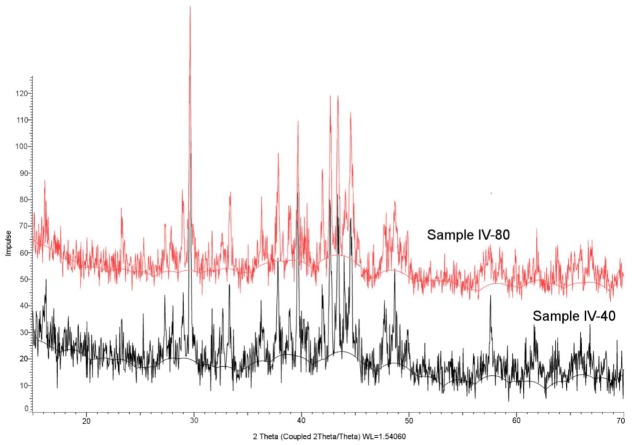
Comparison of XRD patterns for samples IV-40 and IV-80.

**Figure 13 materials-13-00139-f013:**
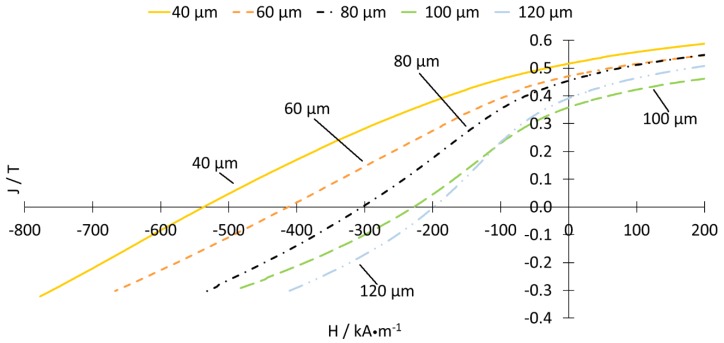
J/H-curves of parameter set IV.

**Figure 14 materials-13-00139-f014:**
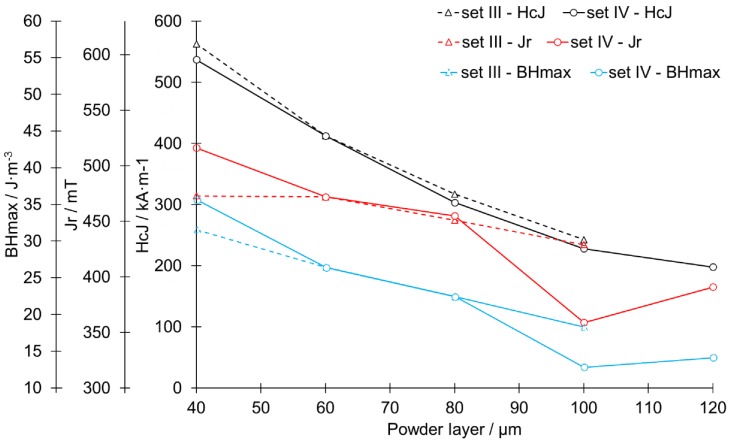
Magnetization test results presented for the function of powder layer thickness.

**Table 1 materials-13-00139-t001:** Description of parameters sets used for experiment.

Parameters Set Code	Power (W)	Laser Speed (mm/s)	Hatching (um)	Beam	Linear Energy (J/mm)
I	40	200	500	defocussed	0.200
II	40	160	500	defocussed	0.250
III	60	200	500	defocussed	0.300
IV	60	160	500	defocussed	0.375

**Table 2 materials-13-00139-t002:** A comparison of a typical grain size of NdFeB magnets manufactured with various methods.

Manufacturing Method	Typical Grain Size
Sintering	10–15 µm [11]
Hot pressing and hot deformation	0.3 µm [26]
LPBF	0.5–3 µm (current study)

**Table 3 materials-13-00139-t003:** Comparison of results with previous studies for Nd7.5Pr0.7Fe75.4Co2.5B8.8Zr2.6Ti2.5.

Feature	Coercivity H_cJ_	Remanence J_r_	Maximum Magnetic Energy Product (BH)max	Relative Density/%	Method
	kA/m	T	kJ/m^3^	%	
This study (max. values)	516 ^a^	0.563 ^b^	35.9 ^c^	90.9 ^d^	LPBF
[19]	775.3	Max. 0.478	39.78	92	LPBF
[17]	N/A	Avr. 0.45 Max. 0.514	N/A	86	LPBF
[18]	915	0.587	52.1	N/A	LPBF
[22] without grain boundary infiltration	520	0.436	N/A	65	LPBF
[22] with grain boundary infiltration with Nd50Tb20Cu30	1207	0.390	N/A	N/A	LPBF
[4]	708.2	0.58	58.1	70	Extrusion
[12]	740	0.310	N/A	N/A	Fused Filament Fabrication
[28]	700	0.52	44	70.5	Injection moulding
Powdered material [27]	670–750	0.73–0.76	80–92	48.4–56.5 ^e^	N/A

^a^ value for set IV, powder layer = 40 μm; ^b^ value for set III, powder layer = 40 μm, ^c^ value for set IV, powder layer = 40 μm; ^d^ value for set IV, powder layer = 60 μm; ^e^ apparent density of the powder.

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
