# Peer review of "Influence of Melt-Pool Stability in 3D Printing of NdFeB Magnets on Density and Magnetic Properties"

_materials, 2019, doi:10.3390/ma13010139_

Round 1

Reviewer 1 Report

Line 71: The detailed procedure of manufacturing the single melt-tracks should be included in the text not only the references. Because, for not familiar reader with this manufacturing procedure the is explanation insufficient.

Line 92: In which atmosphere were the magnets form the spherical powders printed. Was the same Farsoon FS121M printer used for printing the bulk samples and also for single melt- tracks?

Line 88: In figure 3 a and b the scale is missing.

Was the powder during 3d printing in molten state or in semi-solid state?

Did you use any additional heat treatment after 3d printing?

What was the temperature of the substratum (bed) on which the samples were printed?

Actually, the powder the you are using in this study was designed for production of injection molded bonded magnets. Could you explain the drop of coercivity and the max. magnetic energy product of 3d printed magnets made in this study in comparison with bonded magnets made from the same powder.

Did you determined the mechanical properties of LPBF magnets compared to bonded magnets?

What could be the protentional application of LPBF 3D printed magnets?

Reviewer 2 Report

The article entitled Influence of Melt Pool Stability in 3D Printing of NdFeB Magnets on Density and Magnetic Properties is a valuable contribution to the field of additive manufacturing of magnetic materials. The article gives interesting insights on the effects of the fabrication parameters on the magnetic properties via detailed characterization of the melt track. The article separates itself from comparable studies in the field by an in-depth analysis and the variety of characterization techniques used. On the other end, some statements made in the text are unclear or insufficiently supported. In particular, the section from lines 118 to 141 needs revision. The absence of crack analysis is also a problem. For these reasons, I would suggest to accept this paper with minor revision. You will find below more specific comments about the manuscript.

Generally speaking, the text is written with a good English level. However, the article comprises several spelling mistakes and the syntax make the text sometimes difficult to read. Line 44-45: Consider revising this sentence: Additive manufacturing techniques… … functional materials are still under development. It is not clear what is meant by functional materials are still under development. Line 47-48-49: revise syntax. Line 55: more details on the sample shape effect might be valuable here especially in regards to the effect of powder layer thickness describe here. Figure 1: More description on figure 1 is needed. What information can be extracted from figure 1? Line 90-91: The explanation on the radius is not clear. Line 93: incrementally increased... maybe show the step on figure 2. Line 101: I would expect a field in Tesla instead of a voltage. This information is needed to determine the saturation of the magnets. Line 108-109: Are we certain that there is no anisotropy? Either add a reference or an experimental proof to support that comment or edit. As is, it is misleading. Line 118-141: The text is very unclear. I am unsure on what the authors are trying to demonstrate. What is the message conveyed by this section? Why is the length of the melt track not measured? Would it be possible to use the same variables in figures 3 and 5? Please define Lt in the caption of figure 4. Figure 6 and 9: please revise format so that the figures remain readable after printing in black and white. Figure 6: define AR. Line 166-167: Measuring the presence of cracks should be rather straightforward by simple metallographic analysis such as in figure 8. The presence of cracks in a condition is a very fundamental information of great practical interest. Good magnetic properties obtained in a condition with cracks might not be suitable for one application. I would suggest adding more information on crack analysis for each condition. Line 175 and 178: revise the number of significant digits! Line 179-180: Unsupported statement. It is not clear why this implies that the previous layer needs to be melted although it is probably true. Line 194-196: this statement is very incomplete: what conclusions can be made? Line 198-199: not a sentence: revise. Figure 9: visual quality of figure 9 is insufficient for publication. Line 204 to 233: very long text. The real explanation comes with the grain size. Maye reduce the length or present the grain analysis before. Line 304: While it is demonstrated that a decrease in the grain size is obtained by decreasing the layer height, the cause is not explained. Listing plausible hypothesis would add to the value of the paper.
